# LANGUAGE FUSION FOR PARAMETER-EFFICIENT CROSS-LINGUAL TRANSFER

## ABSTRACT

Limited availability of multilingual text corpora for training language models often leads to poor performance on downstream tasks due to undertrained representation spaces for languages other than English. This 'under-representation' has motivated recent cross-lingual transfer methods to leverage the English representation space by e.g. mixing English and 'non-English' tokens at input or extending model parameters to accommodate new languages, which in turn increases computational complexity. To address this, we introduce **F**usion for **La**nguage **Re**presentations (FLARE) in adapters, a method designed to improve both the representation quality and downstream performance for languages other than English. FLARE integrates source and target language representations within the bottlenecks of low-rank LoRA adapters using lightweight linear transformations. This maintains parameter efficiency as the method does not require additional parameters, while improving transfer performance, further narrowing the performance gap to English. Furthermore, the proposed latent representation fusion does not increase the number of input tokens, this way maintaining computational efficiency. Moreover, FLARE provides flexibility to integrate various types of representations, e.g., we show that it is possible to fuse latent translations extracted from machine translation models. A series of experiments across representative cross-lingual natural language understanding tasks, including natural language inference, question-answering and sentiment analysis, demonstrate FLARE's effectiveness, reducing the performance gap to English to 8.39% for XLM-R Large and 12.41% for Llama 3 across our benchmarks, with performance differences averaged over task-specific metrics.[1]

## 1 INTRODUCTION

Representation degradation for 'non-English' languages poses a challenge in the context of pretrained multilingual language models (mPLMs)[2]. Large-scale English text corpora are widely available for self-supervised pretraining, resulting in superior representation quality and downstream task performance when compared to low(er)-resource languages (Lauscher et al., 2020; Yang et al., 2022). Training mPLMs on massively multilingual text data creates a unified representation space that enables cross-lingual information transfer. Despite the substantial improvements, the imbalance in pretraining resources still substantially reduces downstream performance (Winata et al., 2022).

Cross-lingual transfer (termed XLT henceforth) aims to narrow this performance gap by transferring task-specific knowledge acquired in high-resource languages to lower-resource languages (Ruder et al., 2019). Given the dominance of English in pretraining corpora, machine translations (MT) are frequently utilized to avoid processing non-English data (Shi et al., 2010; Artetxe et al., 2020; 2023; Ansell et al., 2023). Techniques utilizing source and target language representation spaces include language mixup (Yang et al., 2022), and concatenating multilingual input sequences for in-context XLT (Kim et al., 2024; Tanwar et al., 2023; Cueva et al., 2024). These approaches, while improving XLT, typically focus on representations in a specific mPLM layer or require extensive training and computational resources by extending the input length. Additionally, these typically

---

[1]Our code repository is available at `https://anonymous.4open.science/r/FLARE-241E`

[2]The domination of the English representation space is observed independent of model architectures, including encoder-only, decoder-only and encoder-decoder transformer (Wu & Dredze, 2020; Lee et al., 2022a; Yang et al., 2022; Wendler et al., 2024; Tang et al., 2024).

rely on high-quality MT output for source language input. Despite the widespread use of discrete machine translations, only few studies explore enhancing the 'internal' information extracted from MT models (Ponti et al., 2021; Schmidt et al., 2024), and MT output is typically not used to model sub-sentential interaction between source and target language representations.

When adapting mPLMs to new tasks and languages, the choice of adaptation method is crucial for downstream performance. Parameter-efficient fine-tuning (PEFT) methods are designed to acquire new knowledge and specialize general-purpose models for specific tasks or domains while minimizing the number of extra parameters required and keeping the large underlying mPLM frozen (Hu et al., 2022). In particular, bottleneck-style adapters extract relevant features from new data by compressing model representations with the assumption that task information can be captured in a lower-dimensional space (Houlsby et al., 2019). This directly aligns with the XLT objectives, providing resource-efficient language and task adaptation capabilities and support for infusing model representations with new knowledge. Similarly, low-rank adapters (LoRA) also create such 'rep-

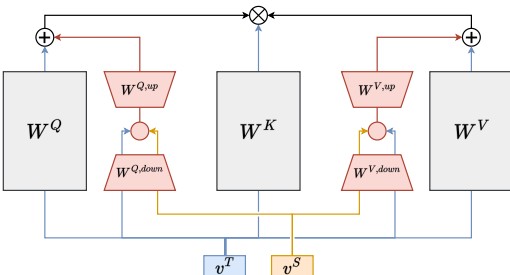

Figure 1: Fusion of source and target representations in LoRA adapters inserted within the query and value matrices. The representations are fused in the adapter bottlenecks and the outputs are added $\oplus$ to the query and value outputs before softmax $\otimes$ activation.

resentation bottlenecks'; they get inserted into the query and value attention modules, and exemplify a widely adopted PEFT approach in large language models (Hu et al., 2022). In XLT, adapters are extensively used for acquiring task and language knowledge (Pfeiffer et al., 2020). Yet, the extent of knowledge transfer within adapters themselves remains underexplored.

In this work, we introduce **F**usion for **La**nguage **Re**presentations (FLARE) *within lower-dimensional adapter bottlenecks* to improve parameter-efficient XLT. As illustrated in Figure 1, we propose *token-wise fusion* of source and target language representations within each transformer block. In contrast to existing methods that leverage source and target representations to improve cross-lingual transfer, our fusion approach maintains computational efficiency by avoiding extending input lengths due to concatenation. Our findings suggest that even *lightweight linear transformations*, such as addition or multiplication, enhance XLT performance, as they allow for the interaction of source and target language representations within the adapter bottlenecks. Besides improved performance, a key advantage of our method lies in its parameter efficiency, as the fusion operations are located within the adapter bottlenecks, thereby not introducing additional parameters while enhancing performance.

Our experiments across natural language inference, sentiment classification, and question answering tasks, using encoder-only, encoder-decoder, and decoder-only mPLMs, demonstrate that our fusion technique effectively reduces the cross-lingual transfer performance gap between English and other languages. For example, FLARE narrows XLM-R Large's average performance gap to English from 9.34% to 8.39% across all evaluated tasks, compared to standard LoRA, with differences averaged over task-specific metrics. Similarly, with decoder-only models like Llama 3, the gap is reduced from 13.63% to 12.41%. Further experiments illustrate that computational efficiency can be further enhanced by using *latent translations* as source language inputs in FLARE, and demonstrate the versatility of the method, which is orthogonal to the choice of mPLMs and MT systems.

**Contributions. 1)** We introduce the FLARE method, fusion for language representations in bottleneck adapters for parameter-efficient cross-lingual transfer. **2)** Our approach effectively narrows the transfer performance gap between English and other languages across various downstream tasks. **3)** We demonstrate the adaptability of our approach by incorporating machine translation encoder representations directly into the mPLM.

## 2 RELATED WORK

**Cross-lingual Representation Transfer.** Enhancing performance for languages underrepresented in the mPLMs' pretraining data often involves aligning and combining representations from various

languages to facilitate XLT (Oh et al., 2022). By concatenating multilingual input sequences, mPLMs leverage a shared representation space across both source and target language inputs (Kim et al., 2024; Tanwar et al., 2023; Cueva et al., 2024). Techniques such as mixtures of task and language adapters have been implemented to merge language representation spaces effectively (Lee et al., 2022b). In projection-based approaches, target language representations are projected onto a high-resource language (e.g., English), to enhance feature extraction in the high-resource language, before re-projecting back to the target language (Xu et al., 2023). Yang et al. (2022) introduced X-Mixup, combining source and target representations in one specific layer of the mPLM using cross-attention. Building on this concept, Cao et al. (2023) used cross-attention with semantic and token-level alignment loss terms, aiming to transfer knowledge from the source to the target language. In contrast, our fusion method modifies the architecture of bottleneck adapters to combine source and target language representations. This enables the efficient fusion of multilingual representations across all transformer layers without adding model parameters, thereby contributing to the stream of *parameter-efficient XLT*.

Representation fusion is also applied to integrate information across different modalities (Fang et al., 2021; Ramnath et al., 2021). For instance, Qu et al. (2024) employed feature routing in cross-modal vision-language tasks, guiding language model representations through the LoRA bottleneck using the last hidden state of a vision model. Our work differs in its scope and fusion methodology: FLARE extracts significantly richer representations from the source and target languages by capturing layer-wise representations for each transformer block in the mPLMs. Moreover, by ensuring dimensional alignment, we perform token-wise representation fusion within adapter bottlenecks, thereby transferring finer-grained information across languages.

**PEFT in Multilingual Language Models and Cross-Lingual Transfer.** PEFT aims to incorporate task or language-specific knowledge into mPLMs without updating all model weights (Pfeiffer et al., 2020). Most prominent techniques include sparse fine-tuning by selectively updating model parameters (Ansell et al., 2022), and inserting adapter modules that reduce trainable parameters to a small fraction of total weights of the underlying mPLM (Houlsby et al., 2019). Furthermore, PEFT modules are composable, and thus information combination from multiple modules is possible (Wang et al., 2022; Lee et al., 2022b). Bottleneck adapters project model representations into a lower-dimensional space and then back to their original dimensions, creating a bottleneck that regulates information flow (Houlsby et al., 2019). During this adaptation process, the weights of the (m)PLM remain frozen. Following the same assumption that task-specific knowledge can be compressed in a low-dimensional space, low-rank adapters (LoRA) (Hu et al., 2022) and its more recent variants (Liu et al., 2024) are widely utilized for fine-tuning language models. They are inserted into the attention modules of transformer architectures, maximizing the capacity to adapt to new task-specific information, while preserving parameter efficiency. In our work, we extend the task and knowledge acquisition capabilities of these adapters by modifying their architecture to process inputs from multiple languages without increasing the parameter count. This involves sharing parameters, such as the adapter projection layers, across language inputs, enabling the fusion of different language representations within the bottleneck (e.g., as implemented in LoRA).

## 3 METHODOLOGY

### 3.1 LANGUAGE REPRESENTATION FUSION

Our methodology is based on the hypothesis that incorporating English with target language representations enhances cross-lingual knowledge transfer and distills task-relevant information into the target language. We assume (MT-created) parallel corpora $\mathcal{P} = \{(x^S, x^T)\}$ during task fine-tuning, where $x$ are instances in the respective source and target language. Our methodology particularly focuses on employing machine-translated 'silver' parallel data, akin to *translate-train* and *translate-test* settings, as we believe this approach is the most realistic in practice. We contend that transferring information during task fine-tuning is more resource-efficient compared to extensive pretraining on large-scale self-supervised text corpora.

Yang et al. (2022) introduced cross-lingual manifold mixup (X-Mixup), aligning multilingual representations within a specific transformer layer using consistency loss terms and a cross-attention module. However, this method introduces additional model parameters and shows performance variability depending on the choice of the mixup layer. Another straightforward and effective method

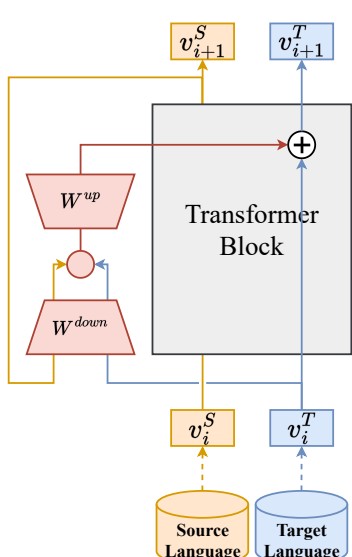

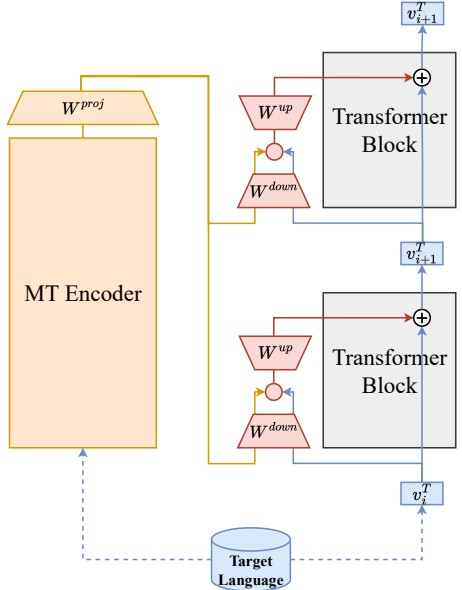

Figure 2: During the forward pass with fusion adapters, source language representations $x^S$ are fused with target language representations $x^T$ in each transformer block $i$. Source representations are extracted by inferencing the mPLM without the fusion adapters.

Figure 3: Illustration of the FLARE MT variant where projected encoder representations from an MT model are directly fused with target language representations within the fusion adapters in the mPLM. Encoder representations from the MT model serve as latent translations, avoiding discretization in the decoder.

for aligning multilingual representations is to concatenate source and target language input sequences $x^{S,T} = [x^S; x^T]$ where $x \in \mathbb{R}^{2m}$, with $m$ representing the sequence length of both source and target languages. This so-called *input-level fusion* enables cross-lingual knowledge transfer across all layers of the mPLM, facilitating in-context learning, which typically does not require additional training (Cueva et al., 2024). However, this approach is computationally expensive due to increased input sequence lengths and encounters scalability issues related to the context length limitations in mPLMs.

To address these limitations, we propose FLARE, a method for *representation-level language fusion* within bottleneck adapters, as illustrated in Figure 1. Instead of extending the input, FLARE processes source and target language representations independently and fuses them only within the adapters, thus preserving computational efficiency. Source language representations $v_i^S$, extracted from the frozen mPLM without adapters, and target language representation $v_i^T$ at transformer block $i$ are down-projected using $W^{down}$ and combined with fusion function $\phi$ (see Section 3.2) to create a fused representation $h = \phi\left(v_{i+1}^S W^{down}, v_i^T W^{down}\right)$, where $h \in \mathbb{R}^{m \times r}$ with sequence length $m$ and bottleneck dimensions $r$. We utilize the source representation $v_{i+1}^S$, which has been processed by the subsequent transformer block, to leverage task-specific information extracted from the source language. Following a standard LoRA procedure, this fused low-rank representation is then up-projected and added to the frozen attention outputs $v^0$ to form the target language output representation $v_{i+1}^T = hW^{up} + v^0$ of the attention block. This enhances the target language adaptation by directing the model's attention to task-relevant information. The down-projection within the bottleneck adapters is applied to both target and source language representations, exploiting the unified embedding space acquired during self-supervised pretraining for cross-lingual adaptation.[3]

---

[3]Assuming that new task information can be learned within low-rank adapters, we posit that task-specific cross-lingual knowledge can be effectively transferred within adapter bottlenecks. This enhances efficiency, and also compresses and aligns task-relevant information, simplifying the complexity of representations $r \ll d$. This setup enables the application of lightweight transformations that merge information from both source and target representations.

A key advantage of representation fusion is the reduction in computational complexity, thereby enhancing parameter efficiency for both task and language adaptation. By processing multilingual inputs separately and only fusing highly compressed representations within adapter bottlenecks, our method avoids the computational overhead associated with quadratic scaling in attention computations for model dimensions $d$, thus enhancing resource efficiency. Furthermore, the memory requirements are limited to the last hidden states obtained from the output of each transformer block.

Moreover, our fusion approach is agnostic regarding the source language representation. This flexibility allows directly leveraging representations extracted from the MT encoder $\mathcal{M}$ as 'latent translations' for fusion. We propose to extract a single representation from the MT model $v^T = \mathcal{M}\left(x^T\right)$, where $v^T \in \mathbb{R}^{m \times d_{\mathcal{M}}}$, which serves as a latent translation. To ensure compatibility between the dimensionality of the MT encoder outputs and the mPLM, we utilize a single linear projection layer $W^{proj}$. This projection is trained during the adaptation to the downstream task in the target language, thereby maintaining efficiency. Moreover, projections between different model representation spaces can be enhanced using self-supervised data, as demonstrated in related studies (Liu et al., 2023). Consequently, the up-projected representation $v^T W^{proj}$ is fused with the target language representation within the adapter bottlenecks of each mPLM layer; see Figure 3. This `FLARE MT` method enhances resource efficiency by bypassing a forward pass in the mPLM, which is required when using discrete text in the source language, and preserves the inherent translation uncertainty within the embeddings by avoiding discretization in the MT decoder, thus mitigating potential translation errors (Ponti et al., 2021; Unanue et al., 2023).

## 3.2 Fusion Functions

To fuse cross-lingual representations in bottleneck adapters, we evaluate both linear and non-linear transformations that do not require additional model parameters, alongside cross-attention. We extract token-wise representations from source and target language sequences, capturing rich contextual information at the token level. Extracting source language and target language representations from the same underlying mPLM ensures matching hidden dimensions $d$ in each transformer layer, facilitating subsequent representation fusion in the low-rank bottleneck adapters.

The down-projected representations in the adapter bottlenecks for source and target languages are denoted as $S = v^S W^{down}$ and $T = v^T W^{down}$, where $S$ and $T$ are representations of dimensions $\mathbb{R}^{m \times r}$. These representations are subsequently combined at the token level through the following fusion functions:

1. element-wise addition (*add*): $S + T$
2. element-wise multiplication (*mul*): $S \circ T$
3. *cross-attention*:[4] $\text{softmax}\left(\frac{W_a^Q S \left(W_a^K T\right)'}{\sqrt{r}}\right) W_a^V T$

$W_a^Q$, $W_a^K$ and $W_a^V$ are the weight matrices of the query, key and value projections in the adapter $a$, respectively, and $'$ denotes the matrix transpose. We focus on lightweight linear transformations to maintain both parameter and computational efficiency.

We extend the linear fusion functions using non-linear transformations through rectified linear units $ReLU\left(S\right)$ and $ReLU\left(T\right)$ (Qu et al., 2024). This addition improves feature extraction capabilities by selectively enabling information flow in token representations. Given the inherent misalignment of multilingual input sequences at the token level, extracting token-level representations for subsequent fusion may introduce alignment issues. We hypothesize that the adapter projections $W^{down}$ aid the alignment of multilingual representations. Further correcting for misalignment between source and target language representations, non-linear transformation functions can restrict propagating misaligned information, which ultimately might improve downstream task performance.

---

[4] Although cross-attention modules add parameters to the adapters, the low bottleneck dimensions $r$, typically smaller than 64, minimize the parameter count in comparison to the model's internal dimensions $d$. Specifically, we utilize a single cross-attention head to maintain efficiency.

## 3.3 TRAINING

For task adaptation in the target language, we insert LoRA adapters into query and value weight matrices of the mPLM previously fine-tuned on English task data (referred to as the *base model*). In FLARE, these adapters implement fusion function $\phi$ that combines source and target language input representations into a single fused representation, as illustrated in Figure 1. Consistent with standard PEFT training, only the task head and LoRA parameters and output layer are trainable, while all other parameters remain frozen.

During the forward pass, detailed in Figure 2, representations from both the source and target languages are extracted at each transformer block. Layer-wise source language representations are obtained from the base model and stacked in matrix $V^S \in \mathbb{R}^{l \times m \times d}$, where $l$ represents the number of layers in the mPLM. Target language representations are obtained during the forward pass through the base model with LoRA adapters. In each layer, source and target language representations are transformed and compressed to lower dimensions $r \ll h$ in the adapter's down-projection $W^{down}$. The shared down-projection layers, applied to both source and target language representations before subsequent fusion, reduce the model's reliance on the English representation space. The final steps include the application of a fusion function and standard up-projection, as already described in Sections 3.1 and 3.2.

## 4 EXPERIMENTAL SETUP

### 4.1 UNDERLYING MODELS AND BASELINES

**mPLMs.** Our experiments are based on various mPLMs including the encoder-only XLM-R Base (270M parameters) and Large (550M) (Conneau et al., 2020), the encoder-decoder mT5-XL (3.7B) (Xue et al., 2021), and the decoder-only Llama 3 (8B) (AI@Meta, 2024).

**Fine-Tuning Setup.** We follow a modular XLT approach where the mPLM is fine-tuned on English task data and subsequently adapted using task data in the target language (Zhao et al., 2021). Unless stated otherwise, models are fine-tuned using $r = 64$ and $\alpha = 128$ in the LoRA configurations, while the hyperparameter configurations of each model are detailed in Table 5 in the appendix.

**Baselines.** We benchmark FLARE against zero-shot cross-lingual transfer, translate-test, and translate-train baselines, X-Mixup, as well as input-level fusion models trained with the same LoRA configurations as the FLARE variants. Model checkpoints are selected on validation data that was machine translated from English to the respective target languages. For FLARE, we select the best performing fusion function for each dataset, with detailed results provided in Table 3. X-Mixup aligns source and target language representations through cross-attention in one specific transformer layer and further aligns model outputs using consistency loss terms (Yang et al., 2022). In contrast, input-level fusion combines source and target language texts directly in the input prompt of the mPLM, doubling the sequence length (Kim et al., 2024; Cueva et al., 2024).[5] More details on the baselines below:

*Zero-Shot XLT.* The base model fine-tuned on English task data is directly evaluated on test data in the target languages without further training.

*Translate-Test.* Test sets in each target language are translated into English using NLLB (NLLB Team et al., 2022). Subsequently, the base model is evaluated on these machine-translated test sets.[6]

*Translate-Train.* The base model is fine-tuned on machine-translated task data in the respective target languages. For each model, fine-tuning is performed with LoRA adapters, establishing strong baselines for the benchmarked XLT approaches. The training data comprises instances translated from English to the target language using NLLB. For fusion methods and X-Mixup, we obtain the required 'silver' parallel data also through MT (using NLLB). The training set consists of parallel sets of English and MT-ed instances, whereas the validation and test sets consist of parallel target language instances and corresponding machine translations into English. We posit that the assumed

---

[5]The context length for input-level fusion models is doubled. Due to memory and context length limitations, these models could not be evaluated for TyDiQA; see later.

[6]Although monolingual English-only PLMs can process machine-translated text, they fail to outperform multilingual models, particularly when evaluating low-resource languages or culturally sensitive content (Ebing & Glavaš, 2024).

absence of gold translations both during training and during inference is the most realistic evaluation of `FLARE` models.

## 4.2 Evaluation Tasks and Datasets

**XNLI** consists of machine-translated sentence pairs that are translated from English to 15 languages (Conneau et al., 2018). The task involves determining whether a sentence entails, contradicts, or is neutral to a given premise.

**NusaX** is a human-annotated sentiment classification dataset that spans 11 Indonesian languages, including low-resource languages (Winata et al., 2023). With 500 labeled instances for each language, the dataset evaluates few-shot adaptation.

**TyDiQA-GoldP** is a human-annotated extractive QA dataset covering 8 languages (Clark et al., 2020). The task is to extract the answer spans from context passages.

Additional information on evaluation languages and datasets used for source language fine-tuning are available in Table 9 in the appendix.

## 4.3 Machine Translations

We utilize NLLB's 3.3B variant (NLLB Team et al., 2022) as the main MT model, with greedy decoding to obtain translations (Artetxe et al., 2023). To ensure consistency in our experimental setup, we also translate languages that are not directly supported by NLLB. Specifically, Madurese (mad) and Ngaju (nij) are translated using the Indonesian language identifier, as these languages are not supported by NLLB[7] (Winata et al., 2023). For translating extractive QA datasets, we enclose the answer spans within marker tokens prior to translation with NLLB (Chen et al., 2023). This method allows us to determine the position of the translated answer spans by locating these marker tokens in the translated text. Instances that fail to retain the answer span marker tokens in the translated output are excluded from evaluation.

## 5 Results and Discussion

**Main Results** displayed in Table 1 confirm our hypothesis that task-specific knowledge can be efficiently transferred from English to other languages within adapter bottlenecks. `FLARE` consistently surpasses the zero-shot, translate-test, and translate-train baselines across various tasks, demonstrating robust performance with machine-translated training data in the target language and machine-translated source language data during inference. Moreover, the results from the few-shot adaptation scenario on NusaX suggest that `FLARE` does not require extensive labeled task data to improve downstream performance on lower-resource languages. While input-level fusion shows competitive results on NusaX, `FLARE` significantly outperforms input-level fusion on XNLI. On average, `FLARE` outperforms input-level fusion by 1.93%, and 1.75% for XLM-R Large and Llama 3, respectively. The results show that input-level fusion replicates English performance, indicating its inability to leverage information from the target language, which highlights a key limitation of this approach (see Table 2). In contrast, `FLARE` mitigates this issue through parameter sharing in the down-projection of the adapter, ensuring that representations from both source and target languages contribute to the final output. Additionally, fusion functions like *add* ensure a balanced combination of both source and target language representations. Beyond performance benefits, `FLARE` reduces the average training time on XNLI by more than 30% when compared to input-level fusion.

Furthermore, `FLARE` consistently outperforms the X-Mixup baselines by 3.10%, and 3.22% for XLM-R Large and Llama 3, respectively. This indicates that `FLARE` mitigates the performance variability observed in X-Mixup by fusing compressed source and target language representations within adapters inserted in all transformer layers.

When comparing `FLARE` with the `FLARE MT` variant which utilizes latent translations, it becomes evident that the mPLM's task-specific source representations enhance downstream performance. In

---

[7]We note that Toba Batak (bbc) is unsupported by NLLB and excluded from the evaluation due to translation artifacts resulting in random classification performance.

Table 1: Average performance (with standard deviation) on natural language understanding datasets. Metrics used are: Accuracy for XNLI, F1 / Exact Match for TyDiQA, and Micro F1 for NusaX. The best-performing results for each XLT model are highlighted in **bold**.[†]

| Model | XNLI | TyDiQA | NusaX | Avg. |
|---|---|---|---|---|
| ***Zero-Shot Cross-Lingual Transfer*** *(models are trained on English data)* | | | | |
| XLM-R Base | $72.65 \pm 0.6$ | $49.08 \pm 1.0 / 37.33 \pm 1.0$ | $62.35 \pm 2.6$ | 59.40 |
| XLM-R Large | $77.42 \pm 0.3$ | $65.21 \pm 0.2 / 54.09 \pm 0.4$ | $75.55 \pm 1.2$ | 70.87 |
| mT5-XL | $78.31 \pm 0.8$ | $64.76 \pm 0.9 / 52.58 \pm 1.6$ | $74.26 \pm 2.0$ | 59.40 |
| Llama 3 8B | $76.86 \pm 0.2$ | $60.26 \pm 1.1 / 45.83 \pm 1.7$ | $51.82 \pm 2.4$ | 60.57 |
| ***Translate-Test*** *(test data is translated to English)* | | | | |
| XLM-R Base | $74.78 \pm 0.4$ | $48.76 \pm 0.8 / 36.94 \pm 1.0$ | $75.93 \pm 0.5$ | 64.52 |
| XLM-R Large | $77.01 \pm 0.1$ | $65.65 \pm 0.2 / 54.19 \pm 0.4$ | $75.41 \pm 0.5$ | 70.87 |
| mT5-XL | $79.13 \pm 0.4$ | $64.88 \pm 0.7 / 52.83 \pm 1.3$ | $75.70 \pm 0.3$ | 71.23 |
| Llama 3 8B | $80.18 \pm 0.4$ | $60.39 \pm 1.1 / 45.85 \pm 1.7$ | $71.99 \pm 1.3$ | 68.43 |
| ***Translate-Train*** *(models are trained on training data translated to the target language)* | | | | |
| XLM-R Base w/ LoRA | $76.95 \pm 0.3$ | $50.06 \pm 0.7 / 37.79 \pm 0.9$ | $70.93 \pm 0.2$ | 63.94 |
| w/ X-Mixup | $69.32 \pm 0.3$ | $44.61 \pm 0.7 / 34.05 \pm 0.8$ | $69.74 \pm 0.7$ | 59.46 |
| w/ input-level fusion | $74.25 \pm 0.3$ | $43.03 \pm 0.2 / 26.81 \pm 0.2$ | $77.40 \pm 1.0$ | 62.19 |
| w/ FLARE MT | $76.49 \pm 0.6$ | $48.99 \pm 1.2 / 37.29 \pm 1.2$ | $71.67 \pm 0.5$ | 63.77 |
| w/ FLARE | $75.51 \pm 0.4$ | $49.96 \pm 0.8 / 37.74 \pm 0.8$ | $72.77 \pm 0.2$ | **64.04** |
| XLM-R Large w/ LoRA | $80.61 \pm 1.0$ | $65.08 \pm 0.9 / 53.83 \pm 1.2$ | $76.77 \pm 1.0$ | 72.28 |
| w/ X-Mixup | $79.51 \pm 1.2$ | $60.68 \pm 1.1 / 51.62 \pm 1.4$ | $74.74 \pm 0.8$ | 70.13 |
| w/ input-level fusion | $77.36 \pm 1.0$ | $55.46 \pm 0.6 / 38.74 \pm 0.3$ | $78.61 \pm 0.3$ | 67.69 |
| w/ FLARE MT | $81.67 \pm 1.2$ | $65.24 \pm 0.5 / 54.00 \pm 0.8$ | $77.16 \pm 0.2$ | 72.82 |
| w/ FLARE | $81.05 \pm 0.4$ | $65.36 \pm 0.6 / 54.35 \pm 0.8$ | $78.78 \pm 0.7$ | **73.23** |
| mT5-XL w/ LoRA | $80.04 \pm 1.6$ | $65.55 \pm 1.0 / 53.72 \pm 1.4$ | $80.45 \pm 0.2$ | 73.38 |
| w/ X-Mixup | $81.16 \pm 1.4$ | $63.84 \pm 1.7 / 49.86 \pm 1.2$ | $78.60 \pm 0.5$ | 72.20 |
| w/ input-level fusion | $79.11 \pm 1.1$ | - | $80.06 \pm 0.3$ | - |
| w/ FLARE MT | $82.76 \pm 1.2$ | - | $80.33 \pm 0.1$ | - |
| w/ FLARE | $81.14 \pm 1.1$ | $65.73 \pm 1.2 / 54.51 \pm 1.5$ | $80.59 \pm 0.2$ | **73.95** |
| Llama 3 8B w/ LoRA | $81.30 \pm 0.4$ | $60.11 \pm 1.5 / 45.28 \pm 1.6$ | $74.14 \pm 1.1$ | 69.38 |
| w/ X-Mixup | $79.25 \pm 0.3$ | $58.03 \pm 1.3 / 43.14 \pm 1.3$ | $72.31 \pm 0.6$ | 67.38 |
| w/ input-level fusion | $78.52 \pm 0.4$ | - | $76.70 \pm 0.5$ | - |
| w/ FLARE MT | $77.37 \pm 0.4$ | - | $73.68 \pm 0.5$ | - |
| w/ FLARE | $81.99 \pm 0.3$ | $60.44 \pm 1.3 / 45.75 \pm 1.2$ | $76.73 \pm 0.9$ | **70.61** |

[†]FLARE results are reported using the fusion functions that yielded the best performance: *add+relu* for XNLI and NusaX, and *mul* for TyDiQA. Results marked with '-' exceed GPU memory or context length limits.

settings where extracting task-specific knowledge for the source representations from the mPLM is challenging, such as when dealing with translation quality issues in lower-resource languages, the richer translation information from the MT model's encoder representations can enhance downstream performance (see Table 6).

**Impact of Translation Quality.**

Translation quality is an important factor when combining source and target language representations. The results in Tables 2 and 1 show that FLARE is robust to lower-quality machine translations while being capable of enhancing performance when gold translations are available. In extractive QA tasks, where lower-quality machine translations negatively impact model performance, FLARE consistently surpasses the translate-train and translate-test baselines. In contrast, the performance of input-level fusion substantially deteriorates when evaluated using machine-translated inputs, underscoring its reliance on the quality of English text inputs. However, when provided with gold translation data, input-level fusion matches or exceeds English performance (see Tables 6, 7, and 8).

Table 2: Average performance for the *translate-train* setting with gold English translations during inference across languages included in the XNLI, and NusaX datasets, representing optimal translation quality. Evaluation metrics include accuracy for XNLI and Micro F1 for NusaX.

| Model | XNLI | NusaX |
|---|---|---|
| *Translate-Train (fusion models are trained on data translated into the target language and evaluated using **gold translations** from the target language to the source language)* | | |
| XLM-R Base | | |
| w/ input-level fusion | 84.63 | 87.87 |
| w/ FLARE | 84.62 | 75.43 |
| XLM-R Large | | |
| w/ input-level fusion | 87.19 | 90.93 |
| w/ FLARE | 88.15 | 84.66 |
| mT5-XL | | |
| w/ input-level fusion | 89.67 | 90.57 |
| w/ FLARE | 86.57 | 80.72 |

Replacing the machine translated source language inputs with gold translations indicates the upper performance limit for XLT models. FLARE benefits significantly from higher-quality translations with its performance scaling directly in line with translation quality. This makes translation accuracy the most influential factor for downstream performance in fusion models.

**On Latent MT Fusion.**

FLARE MT outperforms zero-shot and translate-test baselines and shows competitive performance with regular FLARE. This indicates that errors in discrete translations directly affect downstream performance. In contrast to regular FLARE, the MT encoder representations used in FLARE MT include task-agnostic language information, and therefore do not transfer task knowledge to the target languages. Nonetheless, it provides a

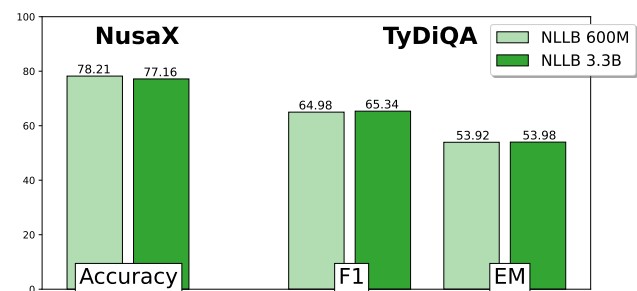

Figure 4: Average performance differences on NusaX and TyDiQA for XLM-R Large using FLARE MT with MT models of different size.

resource-efficient alternative to regular FLARE by avoiding the need for decoding in the MT and eliminating the forward pass in the mPLM, making it especially valuable in scenarios where translation quality is limited. The MT model size, serving as a proxy for translation quality, has a lower impact on performance. The results in Figure 4 show that performance with the NLLB 600M variant is comparable to, or even better than, that with NLLB 3.3B. This suggests that when latent translations from a larger MT model are down-projected to match the model dimensions ($d_{\mathcal{M}} < d_{\mathrm{MT}}$), information is lost, reducing downstream performance.

Additionally, the detailed results for XNLI in Table 6 (appendix) show that FLARE MT is particularly beneficial for lower-resource languages, such as Swahili and Urdu, compared to FLARE, when exposed to large amounts of training data.

**Impact of Fusion Function.**

Table 3 presents the average performance of fusion functions inside LoRAs of XLM-R Large. The results suggest that

Table 3: Average performance of different fusion functions using XLM-R Large with FLARE, evaluated on TyDiQA with F1 / Exact Match and on NusaX with Micro F1.

| Fusion Function | TyDiQA | NusaX |
|---|---|---|
| *Translate-Train (models are trained on data translated to the target language)* | | |
| add | 65.04/53.48 | 79.69 |
| mul | 65.36/54.35 | 78.18 |
| add+relu | 65.06/53.78 | 78.78 |
| cross-attention | 65.04/53.64 | 77.07 |

adding non-linearity to the fusion functions does not provide decisive performance benefits over simpler linear transformations. Notably, the functions *add*, *mul*, and *add+relu* show the best performance. Despite the additional parameters available in cross-attention, the technique does not yield superior downstream performance. This is consistent with the performance of X-Mixup in Table 1. In sum, given that the optimal fusion function appears to be task-dependent, these functions can be regarded as hyperparameters that can also be fine-tuned based on validation data.

**Impact of Adapter Capacity.** Increasing the bottleneck size within LoRA enhances FLARE's performance across datasets, with larger adapter capacities yielding better results. As displayed in 4, even with a small bottleneck size of $r = 8$, FLARE achieves strong performance, demonstrating that highly compressed language representations are sufficient to facilitate cross-lingual transfer in the representation space. However, increasing the adapter capacity further improves performance, particularly for more complex tasks like extractive QA, which require finer-grained representations for optimal fusion. Interestingly, FLARE can leverage larger adapter capacities more effectively compared to regular LoRA adapters without fusion.

Table 4: Average performance for varying adapter bottleneck size $r$ in LoRA; based on XLM-R Large, using FLARE. Evaluation metrics include F1 / Exact Match for TyDiQA and Micro F1 for NusaX.

| Model | $r$ | TyDiQA | NusaX |
|---|---|---|---|
| *Translate-Train (models are trained on training data translated to the target language)* | | | |
| XLM-R Base | 8 | 51.05/38.11 | 63.40 |
| w/ FLARE | | 51.12/39.08 | 66.46 |
| XLM-R Large | | 64.87/53.81 | 77.79 |
| w/ FLARE | | 65.03/53.96 | 79.21 |
| XLM-R Base | 64 | 50.06/37.79 | 70.93 |
| w/ FLARE | | 49.96/37.74 | 72.77 |
| XLM-R Large | | 65.08/53.83 | 76.77 |
| w/ FLARE | | 65.36/54.35 | 78.78 |
| XLM-R Base | 128 | 49.77/37.78 | 70.46 |
| w/ FLARE | | 50.42/38.63 | 73.12 |
| XLM-R Large | | 65.26/53.97 | 77.36 |
| w/ FLARE | | 66.18/55.46 | 79.35 |

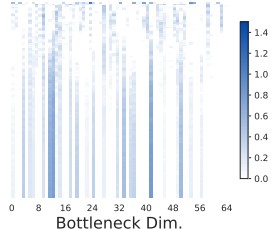
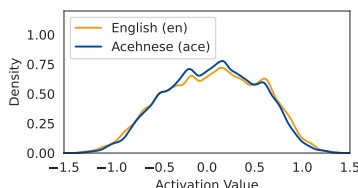

Figure 5: Average activation values for English and Acehnese in the first bottleneck query layer in XLM-R Large for the NusaX test set; *add+relu* fusion.

Figure 6: Average activations in the adapters across all XLM-R Large layers for the NusaX test set.

**Layer-wise Language Activation.**

Figure 6 shows that the magnitudes of source and target language activations across the entire XLM-R Large are comparable. This indicates that `FLARE` does not overly rely on either source or target representations during fusion. Further, Figure 5 displays the average activations for English and Acehnese in the first adapter bottleneck: this confirms that both source and target languages maintain similar activation magnitudes. Hence, subsequent Acehnese representations are infused with the English representations from this initial transfer, integrating balanced source and target language information. Detailed activations for individual instances are illustrated in Figure 7, which show positional activation differences and demonstrate the alignment of source and target languages for information transfer.

## 6 CONCLUSION

We introduced Fusion for Language Representations (`FLARE`), a parameter-efficient method for cross-lingual transfer (XLT) that enhances representation quality and downstream performance for languages other than English. Our experimental results demonstrate that `FLARE` outperforms strong XLT baselines, such as target language fine-tuning with LoRA adapters and input-level fusion, on natural language understanding tasks, effectively narrowing the performance gap with English. `FLARE` is robust to lower-quality machine translations, outperforming strong cross-lingual transfer baselines. A key takeaway is that `FLARE` is representation-agnostic, allowing for the direct integration of latent translations from an MT model in place of translated English text. This further improves resource-efficiency and enhances knowledge transfer for lower-quality translations.

**Limitations.** Our work demonstrates that highly compressed English language representations can be effectively transferred to other languages within adapter bottlenecks. However, our experiments focus on bilingual transfer settings. Extending fusion adapters to integrate multiple target languages is non-trivial, as it requires adapters to extract language-agnostic information across multiple languages.

The proposed `FLARE` method by design relies data availability for both source and target languages. Consequently, the performance of `FLARE` is dependent upon the quality of machine translations, as we also investigated empirically in this work. This dependency poses some significant challenges, particularly for tasks that require precise positional alignment, like extractive question-answering, where the quality of machine translations affects downstream performance and model applicability.

Furthermore, our evaluation exclusively employs English as the high-resource source language for representation fusion. While English is predominantly used in mPLM pretraining corpora, exploring other high-resource languages that share linguistic similarities with the target languages could potentially yield similar or improved cross-lingual transfer performance.

Finally, our choice of base multilingual LMs has been motivated by the current state-of-the-art (SotA) in the field of multilingual NLP and XLT to low-resource languages for NLU tasks. The main models are SotA encoder-only (XLM-R) and encoder-decoder mPLMs (mT5), and decoder-only LLM (Llama 3). However, we note that the LLM technology and its adaptation to XLT for NLU in lower-resource languages has not been proven to be fully mature yet Lin et al. (2024); Razumovskaia et al. (2024).

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

## A  DETAILED EVALUATION RESULTS

**Figure 7** displays average activations within the first adapter bottlenecks in the XLM-R Large model using FLARE and the *add+relu* fusion function. This visualization highlights the positional alignment process between English and Acehnese token representations, with varying activation values across different sequence positions reflecting the dynamics of language representation fusion.

**Table 6** shows the results for the XNLI dataset for each language in zero-shot XLT, translate-test, translate-train settings, including translate-train with gold translations in the source language. The results confirm that `FLARE` consistently improves XTL performance in the translate-train setting across different languages without particular bias towards typological relatedness to English or frequency in pretraining corpora.

**Table 7** details the results for the TyDiQA dataset for each language in the zero-shot XLT, translate-test, and translate-train settings. The outcomes demonstrate that `FLARE` performance extends to tasks including positional information, such as extractive question-answering.

**Table 8** outlines the performance for the NusaX dataset for each language in zero-shot XLT, translate-test, translate-train, and translate-train settings with gold translations in the source language. Even with few training samples, our `FLARE` method demonstrates consistent performance improvements across the low-resource languages included in the NusaX dataset.

## B    TRAINING DETAILS

Our evaluation results are averaged across *three random seeds*. Initially, we fully fine-tune XLM-R Base and XLM-R Large models on English task data. For mT5-XL, fine-tuning is conducted using LoRA adapters set with $r = 64$ and $\alpha = 128$, which are subsequently integrated into the model's weights prior to task fine-tuning in the target languages. Hyperparameter configurations for full-tuning each mPLM are provided in Table 5.

The total computation time for the experimental results exceeds 5,000 GPU hours. All models are trained using half-precision.

## C    PRACTICAL IMPLICATIONS

The practical implementation of bilingual cross-lingual transfer methods, such as FLARE, requires an additional step of language identification to determine bilingual adapter for model inference. While this introduces a preprocessing stage, language identification systems are widely accessible and highly accurate. For example, NLLB achieves a 95% F1 score across 193 FLORES languages, including many low-resource languages (Burchell et al., 2023), ensuring that this step can be seamlessly integrated into real-world applications.

## D    ANOTHER ABLATION: REPRESENTATION FUSION DURING TRAINING ONLY

To investigate the importance of utilizing source language representations during inference, we modified `FLARE` to restrict representation fusion to the training phase only. Specifically, we limited the fusion with source language representations to 50% of the training instances and excluded source language data during inference. This evaluates cross-lingual transfer capabilities based on instance-independent patterns learned from source language representations during training. Our findings reveal that fusion adapters struggle to learn patterns that are independent of specific instances from source language representations during training. As a result, when implemented in the XLM-R Large model on the NusaX test set, the performance of the *train-only* `FLARE` variant decreased by 30%. Crucially, this significant drop underscores the importance of incorporating source language representations during inference to achieve effective cross-lingual adaptation.

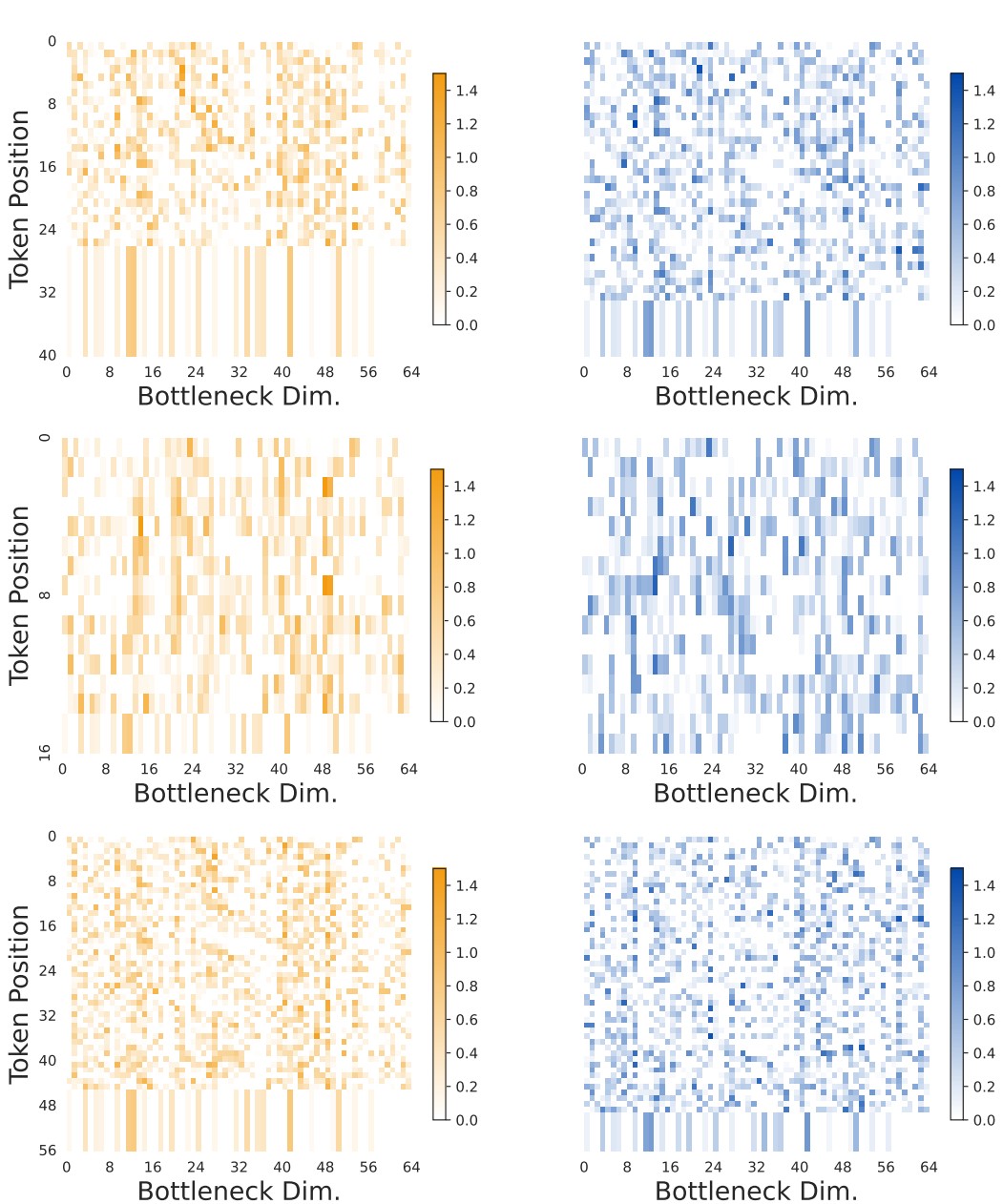

Figure 7: Activation values for individual instances included in the NusaX test set. English and Acehnese activation values are extracted from the first bottleneck query layer in XLMR-Large, which is trained with the *add+relu* fusion function.

Table 5: Hyperparameter configurations for each mPLM across the XNLI, TyDiQA, and NusaX datasets. Values listed in curly braces represent the specific settings used for each dataset in sequential order: {XNLI, TyDiQA, NusaX}.

| Model | Hparam | Values |
|---|---|---|
| XLMR-Base | epochs | 10 |
| | batch size | 32 |
| | sequence length | {128, 512, 128} |
| | learning rate | 2e-5 |
| XLMR-Large | epochs | 10 |
| | batch size | 32 |
| | sequence length | {128, 512, 128} |
| | learning rate | 2e-5 |
| mT5-XL | epochs | 10 |
| | batch size | 64 |
| | sequence length | {128, 512, 128} |
| | learning rate | 2e-4 |
| Llama 3 8B | epochs | 10 |
| | batch size | 64 |
| | sequence length | {128, 512, 128} |
| | learning rate | 2e-4 |

Table 6: Average scores per language in the XNLI dataset. Model performance is evaluated using the Accuracy metric.

| Model | en | ar | bg | de | el | es | fr | hi | ru | sw | th | tr | ur | vi | zh | Avg. |
|---|---|---|---|---|---|---|---|---|---|---|---|---|---|---|---|---|
| *Zero-Shot Cross-lingual Transfer* | | | | | | | | | | | | | | | | |
| XLM-R Base | 84.28 | 71.61 | 77.10 | 75.76 | 74.92 | 78.27 | 77.44 | 69.20 | 74.85 | 63.11 | 71.31 | 71.27 | 65.19 | 74.06 | 73.05 | 72.65 |
| XLM-R Large | 87.81 | 77.32 | 81.84 | 80.70 | 80.91 | 82.92 | 81.89 | 74.00 | 78.88 | 66.06 | 76.45 | 76.36 | 69.77 | 78.20 | 78.54 | 77.42 |
| mT5-XL | 89.04 | 77.50 | 82.81 | 81.72 | 81.56 | 83.93 | 82.93 | 74.91 | 80.39 | 70.64 | 76.39 | 77.38 | 70.86 | 76.85 | 78.50 | 78.31 |
| Llama 3 8B | 92.11 | 76.78 | 80.11 | 83.24 | 80.00 | 86.06 | 84.97 | 72.58 | 81.66 | 57.10 | 72.05 | 77.59 | 63.48 | 80.44 | 79.97 | 76.86 |
| *Translate-Test (translate test data to English using NLLB 3.3B)* | | | | | | | | | | | | | | | | |
| XLM-R Base | 84.28 | 74.42 | 77.64 | 78.35 | 77.90 | 80.05 | 78.44 | 72.56 | 75.80 | 70.29 | 70.65 | 75.92 | 66.38 | 75.64 | 72.94 | 74.78 |
| XLM-R Large | 87.81 | 76.52 | 81.13 | 81.31 | 81.03 | 82.37 | 81.57 | 74.47 | 77.80 | 71.96 | 72.32 | 77.91 | 67.87 | 77.79 | 74.10 | 77.01 |
| mT5-XL | 89.04 | 79.06 | 83.17 | 83.29 | 82.71 | 84.09 | 83.49 | 76.67 | 80.54 | 73.15 | 74.69 | 76.99 | 69.80 | 80.26 | 77.31 | 79.13 |
| Llama 3 8B | 92.11 | 79.75 | 84.70 | 84.82 | 83.95 | 86.38 | 84.94 | 77.37 | 81.31 | 74.60 | 75.03 | 80.99 | 69.59 | 81.01 | 78.07 | 80.18 |
| *Translate-Train (models are trained on training data translated to the target language)* | | | | | | | | | | | | | | | | |
| XLM-R Base | 84.28 | 75.46 | 79.73 | 79.17 | 78.13 | 80.43 | 80.13 | 73.19 | 78.06 | 74.10 | 76.94 | 76.07 | 69.31 | 78.20 | 78.36 | 76.95 |
| w/ X-Mixup | 84.28 | 67.19 | 73.15 | 71.32 | 70.34 | 72.67 | 71.72 | 69.39 | 70.18 | 63.05 | 68.70 | 67.82 | 62.61 | 71.46 | 70.94 | 69.32 |
| w/ input-level fusion | 84.28 | 74.04 | 77.21 | 77.59 | 77.01 | 79.17 | 78.41 | 71.58 | 75.04 | 69.79 | 70.53 | 75.43 | 65.80 | 75.14 | 72.83 | 74.25 |
| w/ FLARE MT | 84.28 | 75.86 | 79.63 | 78.20 | 78.30 | 80.04 | 79.79 | 73.86 | 77.70 | 70.91 | 75.26 | 75.09 | 70.47 | 78.02 | 77.75 | 76.49 |
| w/ FLARE | 84.28 | 75.20 | 77.33 | 78.20 | 76.62 | 78.70 | 78.74 | 72.80 | 75.82 | 70.79 | 75.10 | 75.08 | 68.51 | 77.47 | 76.74 | 75.51 |
| XLM-R Large | 87.81 | 79.46 | 84.25 | 82.77 | 83.34 | 84.62 | 83.18 | 77.91 | 81.54 | 74.56 | 79.85 | 80.84 | 74.84 | 81.81 | 79.56 | 80.61 |
| w/ X-Mixup | 87.81 | 78.46 | 82.44 | 82.00 | 80.22 | 82.85 | 81.38 | 76.07 | 80.58 | 74.13 | 79.40 | 79.20 | 73.65 | 81.50 | 81.24 | 79.51 |
| w/ input-level fusion | 87.81 | 77.40 | 81.52 | 81.19 | 81.47 | 82.52 | 81.18 | 75.00 | 77.90 | 72.24 | 72.88 | 78.58 | 68.57 | 78.22 | 74.30 | 77.36 |
| w/ FLARE MT | 87.81 | 80.98 | 84.83 | 84.19 | 84.04 | 85.10 | 83.87 | 79.11 | 82.14 | 76.78 | 80.39 | 81.77 | 76.36 | 81.95 | 81.83 | 81.67 |
| w/ FLARE | 87.81 | 81.10 | 84.17 | 83.41 | 83.50 | 84.01 | 83.59 | 79.32 | 79.63 | 75.64 | 80.45 | 80.21 | 75.55 | 81.35 | 82.77 | 81.05 |
| mT5-XL | 89.04 | 79.68 | 83.62 | 83.47 | 81.90 | 84.42 | 84.08 | 77.42 | 81.56 | 76.22 | 77.18 | 77.96 | 73.24 | 79.37 | 80.38 | 80.04 |
| w/ X-Mixup | 89.04 | 81.62 | 83.73 | 83.51 | 83.80 | 85.27 | 83.94 | 78.23 | 80.73 | 77.83 | 80.76 | 79.73 | 74.82 | 81.37 | 80.83 | 81.16 |
| w/ input-level fusion | 89.04 | 79.39 | 83.04 | 82.43 | 82.46 | 83.60 | 83.15 | 76.53 | 80.55 | 73.43 | 75.41 | 79.32 | 69.94 | 79.78 | 78.50 | 79.11 |
| w/ FLARE MT | 89.04 | 81.40 | 85.77 | 85.49 | 84.95 | 85.85 | 85.61 | 80.48 | 83.39 | 79.06 | 80.66 | 83.17 | 77.50 | 82.53 | 82.79 | 82.76 |
| w/ FLARE | 89.04 | 81.45 | 83.86 | 83.60 | 82.37 | 85.00 | 83.61 | 79.13 | 81.42 | 77.56 | 79.68 | 80.64 | 74.41 | 81.44 | 81.80 | 81.14 |
| Llama 3 8B | 92.11 | 80.31 | 83.98 | 84.85 | 83.93 | 86.41 | 85.62 | 77.70 | 83.21 | 76.19 | 79.04 | 79.93 | 72.96 | 81.86 | 82.25 | 81.30 |
| w/ X-Mixup | 92.11 | 79.82 | 81.73 | 83.82 | 80.73 | 84.31 | 86.57 | 75.92 | 78.63 | 74.41 | 74.83 | 75.45 | 72.51 | 80.52 | 80.18 | 79.25 |
| w/ input-level fusion | 92.11 | 78.49 | 83.09 | 84.27 | 81.90 | 85.12 | 83.56 | 75.84 | 79.68 | 71.45 | 72.24 | 79.79 | 67.35 | 79.78 | 76.75 | 78.52 |
| w/ FLARE MT | 92.11 | 76.85 | 80.09 | 78.45 | 80.66 | 81.98 | 80.75 | 76.42 | 79.56 | 71.82 | 75.67 | 74.43 | 71.23 | 77.91 | 77.41 | 77.37 |
| w/ FLARE | 92.11 | 78.85 | 83.64 | 84.33 | 85.56 | 86.79 | 87.67 | 78.55 | 83.66 | 78.51 | 79.39 | 80.42 | 76.94 | 81.46 | 82.15 | 81.99 |
| *Translate-Train (fusion models are trained on data translated into the target language and evaluated using gold translations from the target language to the source language)* | | | | | | | | | | | | | | | | |
| XLM-R Base w/ input-level fusion | 84.28 | 84.85 | 84.79 | 84.79 | 84.71 | 84.67 | 84.25 | 84.63 | 84.31 | 84.53 | 84.63 | 84.51 | 84.87 | 84.75 | 84.47 | 84.63 |
| w/ FLARE | 84.28 | 84.63 | 84.63 | 84.53 | 84.67 | 84.55 | 84.57 | 84.35 | 84.39 | 84.65 | 84.87 | 84.87 | 84.79 | 84.67 | 84.49 | 84.62 |
| XLM-R Large w/ input-level fusion | 87.81 | 88.41 | 88.54 | 88.46 | 88.36 | 88.28 | 88.02 | 88.38 | 85.91 | 86.23 | 85.91 | 85.85 | 86.05 | 85.85 | 86.45 | 87.19 |
| w/ FLARE | 87.81 | 88.10 | 88.06 | 88.04 | 88.12 | 88.02 | 88.08 | 88.40 | 88.12 | 88.46 | 88.16 | 88.14 | 88.22 | 88.04 | 88.16 | 88.15 |
| mT5-XL w/ input-level fusion | 89.04 | 90.04 | 89.80 | 89.54 | 89.70 | 89.78 | 89.50 | 89.80 | 89.52 | 89.56 | 89.84 | 89.66 | 89.38 | 89.52 | 89.70 | 89.67 |
| FLARE | 89.04 | 88.62 | 88.74 | 88.80 | 85.34 | 87.83 | 86.19 | 84.31 | 86.12 | 89.66 | 88.49 | 89.56 | 79.22 | 85.33 | 83.73 | 86.57 |

Table 7: Average scores per language in the TyDiQA dataset. Model performance is evaluated using the F1 / Exact Match metrics.

| Model | en | ar | ben | fi | ind | ko | ru | sw | tel | Avg. |
|---|---|---|---|---|---|---|---|---|---|---|
| *Zero-Shot Cross-lingual Transfer* | | | | | | | | | | |
| XLM-R Base | 62.55/52.13 | 51.37/35.58 | 50.62/36.75 | 42.73/29.61 | 58.50/46.65 | 40.60/32.35 | 41.62/31.45 | 58.47/45.54 | 48.73/40.70 | 49.08/37.33 |
| XLM-R Large | 72.50/57.90 | 62.86/51.50 | 72.54/60.48 | 51.57/37.68 | 68.59/58.66 | 53.26/40.48 | 56.57/44.21 | 69.76/59.04 | 86.58/80.68 | 65.21/54.09 |
| mT5-XL | 74.45/66.36 | 56.91/42.34 | 73.78/56.67 | 54.29/39.97 | 68.95/58.30 | 55.60/44.87 | 54.25/41.81 | 69.59/58.03 | 84.68/78.68 | 64.76/52.58 |
| Llama 3 8B | 71.82/65.78 | 59.17/41.72 | 60.20/47.00 | 54.20/34.39 | 64.75/48.93 | 48.93/36.32 | 51.71/35.34 | 66.37/52.90 | 76.78/70.00 | 60.26/45.83 |
| *Translate-Test (translate test data to English using NLLB 3.3B)* | | | | | | | | | | |
| XLM-R Base | 62.55/52.13 | 51.17/36.50 | 47.98/35.39 | 42.59/27.69 | 58.43/47.31 | 40.84/30.58 | 41.72/30.14 | 58.70/47.59 | 48.61/40.30 | 48.76/36.94 |
| XLM-R Large | 72.50/57.90 | 62.62/50.43 | 73.62/60.43 | 52.73/38.64 | 68.86/58.91 | 55.14/41.21 | 55.79/42.73 | 69.68/58.47 | 86.79/82.73 | 65.65/54.19 |
| mT5-XL | 74.19/66.36 | 56.66/42.03 | 73.81/56.67 | 54.09/39.97 | 69.87/59.30 | 55.98/45.30 | 54.66/42.17 | 69.38/58.53 | 84.63/78.68 | 64.88/52.83 |
| Llama 3 8B | 71.82/65.78 | 59.13/41.56 | 60.94/47.00 | 54.02/34.08 | 64.75/48.93 | 48.93/36.32 | 51.71/36.34 | 66.37/52.40 | 77.28/70.13 | 60.39/45.85 |
| *Translate-Train (models are trained on training data translated to the target language)* | | | | | | | | | | |
| XLM-R Base | 62.55/52.13 | 52.46/36.67 | 50.59/38.54 | 44.42/27.37 | 58.90/47.67 | 40.53/29.45 | 42.45/31.52 | 59.50/48.53 | 51.63/42.60 | 50.06/37.79 |
| w/ X-Mixup | 62.93/52.13 | 48.54/32.71 | 47.11/32.55 | 44.05/24.58 | 54.07/46.47 | 34.58/25.34 | 38.46/30.43 | 44.45/43.56 | 45.58/36.73 | 44.61/34.05 |
| w/ FLARE MT | 62.55/52.13 | 51.35/37.38 | 48.52/35.16 | 43.34/28.34 | 59.78/48.41 | 38.47/29.47 | 42.72/31.31 | 58.57/47.61 | 49.22/40.66 | 48.99/37.29 |
| w/ FLARE | 62.55/52.13 | 52.55/37.48 | 49.80/36.75 | 43.67/27.56 | 59.76/48.77 | 40.35/29.65 | 42.28/30.52 | 58.63/47.54 | 52.63/43.67 | 49.96/37.74 |
| XLM-R Large | 72.50/57.90 | 61.51/49.41 | 72.26/58.45 | 51.61/39.51 | 68.50/59.25 | 55.28/41.30 | 55.47/42.58 | 68.53/57.32 | 87.50/82.82 | 65.08/53.83 |
| w/ X-Mixup | 72.50/57.90 | 54.43/47.61 | 68.17/56.17 | 48.30/33.76 | 62.51/58.53 | 49.36/42.47 | 54.82/41.52 | 66.03/52.39 | 81.85/80.55 | 60.68/51.62 |
| w/ FLARE MT | 72.50/57.90 | 61.68/48.87 | 71.46/58.35 | 52.33/37.65 | 68.39/59.80 | 56.37/43.40 | 56.58/43.54 | 68.62/57.60 | 86.55/82.84 | 65.24/54.00 |
| w/ FLARE | 72.50/57.90 | 61.88/49.77 | 71.41/58.55 | 52.55/39.65 | 68.35/60.40 | 56.12/41.64 | 55.48/43.61 | 69.44/57.46 | 87.71/83.71 | 65.36/54.35 |
| mT5-XL | 74.19/66.36 | 59.34/43.28 | 70.71/59.17 | 55.81/40.44 | 70.14/59.45 | 56.92/46.58 | 55.22/40.98 | 70.64/60.24 | 85.59/79.61 | 65.55/53.72 |
| w/ X-Mixup | 74.19/66.36 | 57.95/41.78 | 69.12/60.41 | 54.44/40.59 | 68.47/58.95 | 55.42/28.50 | 52.68/30.60 | 68.92/57.45 | 83.75/80.63 | 63.84/49.86 |
| w/ FLARE | 74.19/66.36 | 58.50/47.66 | 71.99/59.00 | 55.90/40.76 | 70.45/60.98 | 58.12/45.30 | 55.01/42.57 | 69.92/59.90 | 85.95/79.90 | 65.73/54.51 |
| Llama 3 8B | 71.82/64.07 | 58.55/40.94 | 61.51/46.67 | 54.55/35.35 | 64.07/50.15 | 49.59/35.90 | 50.01/33.95 | 64.38/49.83 | 78.22/69.47 | 60.11/45.28 |
| w/ X-Mixup | 71.82/64.07 | 57.73/39.96 | 60.63/44.83 | 53.27/33.60 | 60.92/47.64 | 45.38/30.42 | 48.83/32.65 | 62.29/47.39 | 75.20/68.60 | 58.03/43.14 |
| w/ FLARE | 71.82/64.07 | 59.56/40.63 | 60.97/47.50 | 53.95/35.69 | 63.94/49.13 | 51.18/37.18 | 49.98/33.82 | 65.52/51.71 | 78.38/70.37 | 60.44/45.75 |

Table 8: Average scores per language in the NusaX dataset. Model performance is evaluated using the Micro F1 metric.

| Model | en | ace | ban | bjn | bug | ind | jav | mad | min | nij | sun | Avg. |
|---|---|---|---|---|---|---|---|---|---|---|---|---|
| *Zero-Shot Cross-lingual Transfer* | | | | | | | | | | | | |
| XLM-R Base | 90.50 | 55.31 | 61.60 | 69.51 | 31.94 | 90.44 | 77.40 | 50.26 | 69.81 | 51.30 | 65.93 | 62.35 |
| XLM-R Large | 91.83 | 68.64 | 75.97 | 80.47 | 50.92 | 91.02 | 84.95 | 69.83 | 80.18 | 70.15 | 83.35 | 75.55 |
| mT5-XL | 91.38 | 72.43 | 76.38 | 79.75 | 44.84 | 90.61 | 87.46 | 61.38 | 77.76 | 65.27 | 86.73 | 74.26 |
| Llama 3 8B | 88.98 | 47.60 | 52.18 | 51.66 | 40.90 | 73.33 | 55.04 | 51.84 | 52.88 | 46.09 | 46.64 | 51.82 |
| *Translate-Test (translate test data to English using NLLB 3.3B)* | | | | | | | | | | | | |
| XLM-R Base | 90.50 | 76.53 | 75.76 | 84.33 | 72.14 | 86.22 | 83.83 | 56.09 | 81.61 | 58.32 | 84.47 | 75.93 |
| XLM-R Large | 91.83 | 76.30 | 72.70 | 83.39 | 70.33 | 86.78 | 83.28 | 58.99 | 81.04 | 57.03 | 84.32 | 75.41 |
| mT5-XL | 91.38 | 76.25 | 73.43 | 81.68 | 69.25 | 86.64 | 83.50 | 60.63 | 82.43 | 60.49 | 82.68 | 75.70 |
| Llama 3 8B | 88.98 | 72.12 | 75.53 | 78.20 | 66.55 | 81.02 | 78.44 | 58.26 | 77.36 | 55.63 | 76.82 | 71.99 |
| *Translate-Train (models are trained on training data translated to the target language)* | | | | | | | | | | | | |
| XLM-R Base | 90.50 | 69.90 | 70.13 | 77.99 | 62.13 | 86.95 | 80.60 | 51.94 | 78.30 | 53.93 | 77.45 | 70.93 |
| w/ X-Mixup | 90.50 | 65.82 | 69.33 | 71.51 | 70.70 | 86.60 | 81.38 | 52.00 | 75.21 | 56.48 | 68.32 | 69.74 |
| w/ input-level fusion | 90.50 | 80.24 | 74.69 | 83.24 | 71.89 | 89.63 | 84.87 | 61.50 | 83.27 | 59.39 | 85.29 | 77.40 |
| w/ FLARE MT | 90.50 | 70.28 | 67.88 | 77.87 | 62.51 | 89.90 | 81.11 | 56.85 | 79.61 | 55.24 | 75.47 | 71.67 |
| w/ FLARE | 90.50 | 69.81 | 71.68 | 76.89 | 70.40 | 87.34 | 79.55 | 58.15 | 79.60 | 58.21 | 76.05 | 72.77 |
| XLM-R Large | 91.83 | 75.14 | 73.75 | 81.90 | 61.12 | 89.48 | 85.60 | 69.56 | 81.47 | 65.45 | 84.20 | 76.77 |
| w/ X-Mixup | 91.83 | 70.28 | 72.75 | 80.39 | 58.46 | 87.82 | 84.85 | 62.06 | 81.08 | 68.08 | 81.61 | 74.74 |
| w/ input-level fusion | 91.83 | 77.84 | 76.48 | 83.12 | 69.37 | 89.47 | 87.16 | 66.95 | 80.17 | 68.86 | 86.64 | 78.61 |
| w/ FLARE MT | 91.83 | 73.52 | 75.45 | 80.75 | 58.68 | 90.95 | 86.59 | 69.00 | 83.82 | 68.13 | 84.76 | 77.16 |
| w/ FLARE | 91.83 | 75.32 | 76.83 | 80.91 | 70.23 | 90.17 | 87.21 | 70.67 | 85.07 | 69.21 | 82.18 | 78.78 |
| mT5-XL | 91.38 | 80.52 | 81.68 | 85.66 | 65.67 | 89.73 | 90.28 | 70.54 | 82.64 | 69.20 | 88.64 | 80.45 |
| w/ X-Mixup | 91.38 | 80.33 | 74.75 | 86.78 | 68.78 | 88.33 | 88.58 | 68.52 | 83.45 | 65.73 | 83.94 | 78.60 |
| w/ input-level fusion | 91.38 | 80.99 | 79.25 | 84.88 | 71.55 | 89.63 | 86.64 | 67.03 | 83.35 | 68.80 | 88.47 | 80.06 |
| w/ FLARE MT | 91.38 | 81.18 | 83.44 | 84.91 | 66.10 | 90.01 | 89.68 | 70.64 | 84.82 | 71.79 | 88.71 | 80.33 |
| w/ FLARE | 91.38 | 81.72 | 81.66 | 85.42 | 66.39 | 89.24 | 89.98 | 70.06 | 84.06 | 69.31 | 88.03 | 80.59 |
| Llama 3 8B | 88.98 | 75.47 | 72.50 | 81.29 | 63.80 | 87.00 | 80.81 | 63.87 | 78.49 | 61.37 | 76.87 | 74.14 |
| w/ X-Mixup | 88.98 | 75.32 | 63.19 | 80.72 | 70.74 | 87.16 | 80.61 | 65.09 | 73.93 | 55.89 | 70.44 | 72.31 |
| w/ input-level fusion | 88.98 | 74.88 | 75.18 | 84.00 | 67.54 | 89.67 | 83.65 | 65.66 | 82.05 | 62.88 | 81.51 | 76.70 |
| w/ FLARE MT | 88.98 | 70.42 | 72.22 | 77.91 | 61.33 | 89.11 | 82.25 | 68.83 | 78.46 | 61.95 | 74.30 | 73.68 |
| w/ FLARE | 88.98 | 76.15 | 75.87 | 81.94 | 69.72 | 88.22 | 83.33 | 65.59 | 82.11 | 62.19 | 82.19 | 76.73 |
| *Translate-Train (fusion models are trained on data translated into the target language and evaluated using gold translations from the target language to the source language)* | | | | | | | | | | | | |
| XLM-R Base w/ input-level fusion | 90.50 | 90.04 | 88.45 | 88.23 | 88.51 | 89.83 | 90.19 | 84.76 | 88.55 | 82.79 | 87.38 | 87.87 |
| w/ FLARE | 90.50 | 74.79 | 77.66 | 79.66 | 85.84 | 89.52 | 82.06 | 51.11 | 80.25 | 57.10 | 76.31 | 75.43 |
| XLM-R Large w/ input-level fusion | 91.83 | 91.24 | 91.08 | 90.55 | 90.69 | 91.99 | 90.88 | 91.23 | 91.07 | 90.07 | 90.52 | 90.93 |
| w/ FLARE | 91.83 | 89.24 | 88.98 | 82.55 | 90.07 | 90.22 | 88.15 | 71.20 | 87.58 | 72.93 | 85.71 | 84.66 |
| mT5-XL w/ input-level fusion | 91.38 | 91.39 | 90.39 | 91.47 | 91.54 | 90.88 | 89.49 | 88.87 | 90.86 | 89.20 | 91.60 | 90.57 |
| w/ FLARE | 91.38 | 83.80 | 80.55 | 84.06 | 64.70 | 88.32 | 90.50 | 74.36 | 83.64 | 69.29 | 88.00 | 80.72 |

Table 9: Overview of languages and corresponding source data used in the experiments, categorized by task.

| Task | Language | ISO Code | Source |
|---|---|---|---|
| XNLI | Arabic | ar | |
| | Bulgarian | bg | |
| | Chinese | zh | |
| | French | fr | |
| | German | de | |
| | Greek | el | |
| | Hindi | hi | Crowd-sourced (Williams et al., 2018) |
| | Russian | ru | |
| | Spanish | es | |
| | Swahili | sw | |
| | Thai | th | |
| | Turkish | tr | |
| | Urdu | ur | |
| | Vietnamese | vi | |
| TyDiQA | Arabic | ar | |
| | Bengali | ben | |
| | Finnish | fi | |
| | Indonesian | ind | Wikipedia (Clark et al., 2020) |
| | Korean | ko | |
| | Russian | ru | |
| | Swahili | sw | |
| | Telugu | tel | |
| NusaX | Acehnese | ace | |
| | Balinese | ban | |
| | Banjarese | bjn | |
| | Buginese | bug | |
| | Indonesian | ind | SmSA (Purwarianti & Crisdayanti, 2019) |
| | Javanese | jav | |
| | Madurese | mad | |
| | Minangkabau | min | |
| | Ngaju | nij | |

