# OpenReview forum: "Language Fusion for Parameter-Efficient Cross-lingual Transfer"
_ICLR.cc/2025/Conference — ICLR 2025 Conference Withdrawn Submission_

### Official Review · Reviewer_juGF · 2024-10-27

**Soundness:** 2
**Presentation:** 2
**Contribution:** 2
**Rating:** 3
**Confidence:** 4

**Summary:**

This paper introduces an adapter-based fusion technique to enhance the performance of multilingual language models on non-English languages. The proposed method improves cross-lingual transfer (XLT) with minimal additional computational cost and can leverage the hidden states of pre-trained translation models.

**Strengths:**

The approach demonstrates broad applicability across various models, as evidenced by experiments conducted on XLM-R, mT5, and Llama3. Additionally, the integration of adapters with cross-lingual transfer strategies makes the method cost-efficient.

**Weaknesses:**

Firstly, the modification to LoRA appears similar to existing methods [1, 2], and the observed improvements in XLT performance following adapter fine-tuning with representations from both source and target languages are unsurprising. According to the results presented in Table 1, the enhancements achieved by FLARE are marginal and constrained by the underlying translation model's performance, limiting its effectiveness for low-resource languages.

Secondly, the chosen baselines seem weak. For instance, X-Mixup consistently degrades the original model’s performance after fine-tuning. Moreover, the input-level fusion baseline sometimes outperforms FLARE on the NusaX dataset, and its performance on TyDiQA is not reported due to out-of-memory issues. Given that TyDiQA does not typically involve long-context scenarios, the omission of input-level fusion results for this task is problematic.

Furthermore, the manuscript requires significant improvements in clarity and presentation:

- Figure Illustrations: Figures 2 and 3 are confusing. For example, it is unclear why the target language is fed into the MT Encoder in figure 3. The proposed method essentially involves down-projecting and fusing representations from both source and target languages (whether from raw tokens or pre-trained MT encoder states) before up-projecting them to the decoder's next layer. However, the methodological descriptions are difficult to follow.

- Training Objective: The paper does not clearly specify the training objective used for updating the adapter module. It is unclear whether the module is fine-tuned using a classifier head for tasks like XNLI or if it employs a self-supervised objective such as masked language modeling, as used in XLM-R. Clarification of the training objective in Section 4.1 is necessary.

Overall, while the method shows potential, the marginal performance gains, weak baseline comparisons, and lack of clarity in methodology and presentation diminish the paper’s contribution.

[1] Zhao et al., (2024). AdaMergeX: Cross-Lingual Transfer with Large Language Models via Adaptive Adapter Merging

[2] Xu et al., (2023). Condensing Multilingual Knowledge with Lightweight Language-Specific Modules

**Questions:**

Can you give more intuition/explanation for why FLARE-MT does not work on Llama3 model (I see a drop in both XNLI and NusaX)?

Comparing FLARE-MT with FLARE, it seems that their improvement over different task/language is different. But it seems that for high-resource languages, directly using discrete token (FLARE) works the best. The paper mentioned that for language like Urdu, FLARE-MT gives more improvement. But at the same time, Llama3 + FLARE seems to work much better than FLARE-MT on these languages (see Table 6 ur column). Can you give some explanation for why FLARE-MT works or does not work in these scenarios?

---

> ### Author Response · Authors · 2024-11-23
> **Author Response to Official Review of Submission11107 by Reviewer juGF**
>
> > Q1: Firstly, the modification to LoRA appears similar to existing methods [1, 2], and the observed improvements in XLT performance following adapter fine-tuning with representations from both source and target languages are unsurprising. According to the results presented in Table 1, the enhancements achieved by FLARE are marginal and constrained by the underlying translation model's performance, limiting its effectiveness for low-resource languages.
>
> We disagree with this assessment. FLARE introduces a novel modification to bottleneck adapters like LoRA by enabling them to process multiple inputs, such as source and target language representations, within a single adapter. We employ a fusion function to combine the two representations in the adapter bottleneck to output a single fused representation.
>
> The AdaMergeX approach [1], by contrast, **does not modify the adapter architecture** but instead trains separate LoRA adapters for each target language, which are subsequently merged. Similarly, the LMS approach [2] utilizes a mixture of experts architecture for machine translation and **does not merge representations from different languages nor does it modify the LoRA adapter architecture**. Additionally, since LMS is designed specifically for machine translation, it does not incorporate the machine translated target language as an input, as this serves as the prediction target.
>
> In Section 5, we examine the effect of translation quality, including on low-resource languages, and find that FLARE maintains robustness to lower-quality translations, outperforming regular LoRA and other baselines. Additionally, the consistent performance gains achieved on NLU tasks by FLARE align with improvements reported in related cross-lingual transfer studies, such as [1,3].
>
> References:
>
> - [1] Zhao et al., (2024). AdaMergeX: Cross-Lingual Transfer with Large Language Models via Adaptive Adapter Merging
> - [2] Xu et al., (2023). Condensing Multilingual Knowledge with Lightweight Language-Specific Modules
> - [3] Huiyun Yang, Huadong Chen, Hao Zhou, and Lei Li. Enhancing cross-lingual transfer by manifold mixup. In International Conference on Learning Representations, 2022. URL https://openreview.net/forum?id=OjPmfr9GkVv.
>
> > Q2: Secondly, the chosen baselines seem weak. For instance, X-Mixup consistently degrades the original model’s performance after fine-tuning. Moreover, the input-level fusion baseline sometimes outperforms FLARE on the NusaX dataset, and its performance on TyDiQA is not reported due to out-of-memory issues. Given that TyDiQA does not typically involve long-context scenarios, the omission of input-level fusion results for this task is problematic.
>
> We believe this comment is based on a misunderstanding. While X-Mixup consistently underperforms compared to other strong baselines in Table 1, it does improve performance over zero-shot baselines on both the XNLI and NusaX datasets. Additionally, the translate-train performance reported in the first row of each model already includes fine-tuning on machine-translated training data with LoRA adapters. This provides a strong XLT baseline, as discussed in Sections 4.1 and 5. We believe this clarification also addresses the performance concerns raised in Q1.
>
> For TyDiQA, we use a sequence length of 512 to fit full text inputs across all languages. Extending the sequence length to 1024 exceeds XLM-R's capacity and result in memory issues for the LLaMA and mT5 models. Nevertheless, we benchmark input-level fusion with truncated input sequences.
> The results below demonstrate that FLARE significantly outperforms input-level fusion on TyDiQA, achieving an improvement of 6.93% (F1) and 10.93% (EM) for XLM-R, and 9.90% (F1) and 15.61% (EM) for XLM-R Large. Across all benchmarked datasets, FLARE achieves an average performance gain of 1.85% and 5.54% over input-level fusion for XLM-R and XLM-R Large, respectively.
>
> | Model                             | XNLI  | TyDiQA         | NusaX | Avg.  |
> |-----------------------------------|-------|----------------|-------|-------|
> | XLM-R w/ LoRA                     | 76.95 | 50.06 / 37.79  | 70.93 | 63.94 |
> | XLM-R w/ input-level fusion       | 74.25 | 43.03 / 26.81  | 77.4  | 62.19 |
> | XLM-R w/ FLARE                    | 75.51 | 49.96 / 37.74  | 72.77 | **64.04** |
> |                                   |       |                |       |       |
> | XLM-R Large w/ LoRA               | 80.61 | 65.08 / 53.83  | 76.77 | 72.28 |
> | XLM-R Large w/ input-level fusion | 77.36 | 55.46 / 38.74  | 78.61 | 67.69 |
> | XLM-R Large w/ FLARE              | 81.05 | 65.36 / 54.35  | 78.78 | **73.23** |
>
> **Actions taken**: We have updated Table 1 by adding the "w/ LoRA" suffix to translate-train model entries to explicitly indicate the use of LoRA adapters. We included the performance results for input-level fusion on the TyDiQA dataset.

---

> > ### Author Response · Authors · 2024-11-23
> > **Author Response to Official Review of Submission11107 by Reviewer juGF (continued)**
> >
> > > Q3: Figure Illustrations: Figures 2 and 3 are confusing. For example, it is unclear why the target language is fed into the MT Encoder in figure 3. The proposed method essentially involves down-projecting and fusing representations from both source and target languages (whether from raw tokens or pre-trained MT encoder states) before up-projecting them to the decoder's next layer. However, the methodological descriptions are difficult to follow.
> >
> > FLARE MT utilizes the MT encoder to create a latent translation representation (v^S) of the target language input. A key distinction here is that the source language text is not needed for FLARE MT. The process involves feeding the target language input into the MT encoder, which creates representation v^S by encoding the text as if it were being translated to English. Importantly, this process doesn’t generate actual English text, but rather extracts the encoder’s internal representation.
> > This latent translation representation (v^S) serves a similar purpose to the source language representation shown in Figure 2. However, there is an important difference, as v^S remains constant across transformer blocks, as it is not processed by the frozen mPLM. This is why we denote it as v^S rather than v^S_i (where i would indicate the transformer block number).
> >
> > We kindly ask for specific feedback about unclear aspects of the methodology description in Section 3.1 to help us improve its clarity further.
> >
> > > Q4: Training Objective: The paper does not clearly specify the training objective used for updating the adapter module. It is unclear whether the module is fine-tuned using a classifier head for tasks like XNLI or if it employs a self-supervised objective such as masked language modeling, as used in XLM-R. Clarification of the training objective in Section 4.1 is necessary.
> >
> > The training details are described in lines 264-272, where we explain that FLARE employs supervised task adaptation using specific task heads. For our evaluation tasks, these include classification heads for NLI, sentiment analysis, and question answering tasks.
> > In Section 4.1, under the "Fine-tuning Setup" paragraph, we provide a comprehensive description of our cross-lingual transfer approach, focusing on task-specific fine-tuning. Furthermore, we thoroughly detail FLARE's application during downstream task fine-tuning in both Sections 3.1 and 5.
> >
> > > Q5: Can you give more intuition/explanation for why FLARE-MT does not work on Llama3 model (I see a drop in both XNLI and NusaX)?
> >
> > Table 1 demonstrates that FLARE-MT does improve upon zero-shot performance for Llama 3. However, its performance relative to FLARE and regular LoRA is lower, which we attribute to LLaMA's larger architecture (32 layers with 4096 model dimensions) compared to models like mT5-xl (24 layers with 2048 model dimensions). We hypothesize that the projection (W^{proj}) from the MT to Llama can be improved by: (i)  increasing the number of training steps, (ii) Extending the projection layer to an MLP [1], or (iii) adding a self-supervised training step for the projection, similar to LLaVA [1].
> >
> > References:
> >
> > - [1] Liu, H., Li, C., Wu, Q., & Lee, Y. J. (2023). Visual Instruction Tuning. Thirty-Seventh Conference on Neural Information Processing Systems. URL https://openreview.net/forum?id=w0H2xGHlkw
> >
> > > Q6: Comparing FLARE-MT with FLARE, it seems that their improvement over different task/language is different. But it seems that for high-resource languages, directly using discrete token (FLARE) works the best. The paper mentioned that for language like Urdu, FLARE-MT gives more improvement. But at the same time, Llama3 + FLARE seems to work much better than FLARE-MT on these languages (see Table 6 ur column). Can you give some explanation for why FLARE-MT works or does not work in these scenarios?
> >
> > Following on the description of suggested improvements for the FLARE MT method for Llama 3 in Q5, we focus on the explanations for FLARE vs. FLARE MT for the XLM-R models and mT5. For XLM-R and mT5 models, FLARE-MT's effectiveness stems from the MT encoder's ability to capture rich latent translations while preserving translation uncertainty. This uncertainty information is particularly valuable for low-resource languages where translation errors occur more frequently, but is absent when using translated English text in the standard FLARE approach.  It is important to note that FLARE-MT's effectiveness depends on sufficient training data availability, as evidenced by the performance differences between XNLI and NusaX datasets.
> >
> > **Given our detailed clarifications addressing the reviewers misunderstanding in Q1 and FLARE's methodology, baseline comparisons, and performance analysis across different language scenarios, we kindly ask the reviewer to reconsider the rating score.**

---

> > > ### Comment · Reviewer_juGF · 2024-11-25
> > > **Official Comment by Reviewer juGF**
> > >
> > > Thank the author for providing additional clarification and experiments. I will maintain my score and do not advocate for acceptance.
> > >
> > > My major complain is still on novelty: I don't think adding/multiplying/cross-attention on down-projected representations of source and target language is novel.
> > >
> > > On experiments:
> > >
> > > > While X-Mixup consistently underperforms compared to other strong baselines in Table 1, it does improve performance over zero-shot baselines on both the XNLI and NusaX datasets. Additionally, the translate-train performance reported in the first row of each model already includes fine-tuning on machine-translated training data with LoRA adapters.
> > >
> > > - Since X-mixup is trained to align source and target representation, it should definitely outperform zero-shot baseline. But it cannot outperform a baseline with LoRA fine-tuninig. Same issue holds for the other input-level fusion baseline.
> > >
> > > - Model fine-tuned on machine translated training data with LoRA should be the least challenging method to beat, since your method also use LoRA fine-tuning and involve fusion on representation level.  However, the performance gain in Table 1 is mixed and marginal.

---

### Official Review · Reviewer_EFqY · 2024-10-28

**Soundness:** 3
**Presentation:** 2
**Contribution:** 2
**Rating:** 5
**Confidence:** 4

**Summary:**

The authors present a method called Fusion for Language Representations (FLARE) to enhance cross-lingual transfer in multilingual language models. FLARE integrates source and target language representations within LoRA adapters, aiming to improve performance on downstream tasks for languages other than English. Experimental results across tasks such as natural language inference, question answering, and sentiment analysis demonstrate that FLARE effectively reduces the performance gap between English and other languages. The method shows consistent performance gains across various model architectures, including XLM-R, mT5, and Llama 3.

**Strengths:**

1. FLARE enhances cross-lingual transfer without increasing the model's parameter count or computational overhead. By integrating source (e.g., English) and target language representations within LoRA adapters, it maintains efficiency while improving performance.
2. The method demonstrates consistent improvements across various tasks—natural language inference, question answering, and sentiment analysis, highlighting its general applicability in cross-lingual settings.
3. FLARE is evaluated on multiple model architectures (XLM-R, mT5, Llama 3), showing its effectiveness across encoder-only, encoder-decoder, and decoder-only models.

**Weaknesses:**

1. The main contribution appears to be the integration of source and target language representations within LoRA adapters, which may be seen as an incremental extension of existing methods. The paper could benefit from a clearer articulation of how FLARE differentiates itself from prior work and what specific novel insights it brings to the field.

2. While FLARE shows consistent improvements, the performance gains over baselines are relatively modest. A more thorough analysis is needed to demonstrate the practical significance of these gains and to justify the method's effectiveness compared to the simplicity of the approach.

3. Section 3.2, which introduces the fusion functions, is somewhat hard to follow. It is unclear which of the three fusion functions (addition, multiplication, cross-attention) is primarily used in the experiments. Providing more detailed explanations and justifications for the choice of fusion functions would improve the clarity of the method.

**Questions:**

1. It seems that FLARE MT does not require the input in the source language during inference. Could the authors explain why this approach works effectively without the source input? Additionally, what would be the impact of feeding the source input into the MT encoder for this method?

---

> ### Author Response · Authors · 2024-11-23
> **Author Response to Official Review of Submission11107 by Reviewer EFqY**
>
> > Q1: The main contribution appears to be the integration of source and target language representations within LoRA adapters, which may be seen as an incremental extension of existing methods. The paper could benefit from a clearer articulation of how FLARE differentiates itself from prior work and what specific novel insights it brings to the field.
>
> We disagree with the premise of this remark. The fusion of source and target language representations for cross-lingual transfer is a novel contribution. By fusing these representations within bottleneck adapters like LoRA, FLARE achieves both parameter efficiency and performance gains. While we utilize LoRA adapters, our approach modifies their architecture to combine multiple language representations, which is different from their original use case. This represents a significant methodological contribution rather than an incremental extension.
>
> **Actions taken**: We have extended our explanation in Section 2 to clarify how FLARE differentiates itself from prior work.
>
> > Q2: While FLARE shows consistent improvements, the performance gains over baselines are relatively modest. A more thorough analysis is needed to demonstrate the practical significance of these gains and to justify the method's effectiveness compared to the simplicity of the approach.
>
> Our findings highlight that compressed language representations can be effectively merged within adapters, which is a novel and valuable contribution. Compared to the strong input-level fusion baseline, FLARE improves average performance by 1.93% and 1.75% on XLM-R large and LLaMA 3, respectively. These gains align with performance improvements reported in recent studies on cross-lingual transfer for NLU tasks [1,2], reinforcing both the research and practical significance of FLARE.
>
> References:
>
> - [1] Huiyun Yang, Huadong Chen, Hao Zhou, and Lei Li. Enhancing cross-lingual transfer by manifold mixup. In International Conference on Learning Representations, 2022. URL https://openreview.net/forum?id=OjPmfr9GkVv.
> - [2] Zhao, Y., Zhang, W., Wang, H., Kawaguchi, K., & Bing, L. (2024). AdaMergeX: Cross-Lingual Transfer with Large Language Models via Adaptive Adapter Merging. arXiv [Cs.CL]. Retrieved from http://arxiv.org/abs/2402.18913
>
> **Actions taken**: We have highlighted these findings, along with the detailed results presented in Tables 6, 7, and 8.
>
> > Q3: Section 3.2, which introduces the fusion functions, is somewhat hard to follow. It is unclear which of the three fusion functions (addition, multiplication, cross-attention) is primarily used in the experiments. Providing more detailed explanations and justifications for the choice of fusion functions would improve the clarity of the method.
>
> As highlighted in the first paragraph of Section 3.2, we evaluate multiple fusion functions, providing insights into the information that is fused within the adapters. Our results, detailed in Table 1, show that we use the add+relu fusion function for the XNLI and NusaX datasets, while the mul fusion function is used for TyDiQA. Section 5 further provides a performance overview of different fusion functions, demonstrating that tuning the fusion function as a hyperparameter can yield additional performance gains.
>
> **Actions taken**:  We have included additional clarifications on the fusion functions in the description of the main results in Section 5.
>
> > Q4: It seems that FLARE MT does not require the input in the source language during inference. Could the authors explain why this approach works effectively without the source input? Additionally, what would be the impact of feeding the source input into the MT encoder for this method?
>
> FLARE MT utilizes the MT encoder to generate a latent translation from the target language input, so only the target language text is needed as input. Feeding a (machine) translated source input to the MT encoder would essentially instruct it to translate English text into English. This process is unlikely to yield a representation that enhances cross-lingual transfer performance, as it would not introduce any additional linguistic information beneficial to the model.
>
> **Given the explanations provided and considering the reviewer’s acknowledgment of FLARE’s consistent performance gains for natural language understanding tasks, alongside its general applicability and parameter efficiency, we kindly request the reviewer to raise the rating score.**

---

> > ### Comment · Reviewer_EFqY · 2024-11-26
> >
> > Thank you for your efforts in addressing my concerns. After carefully reviewing your response, I decide to keep my score as it is.

---

### Official Review · Reviewer_xTkb · 2024-11-04

**Soundness:** 3
**Presentation:** 3
**Contribution:** 3
**Rating:** 6
**Confidence:** 3

**Summary:**

**Paper Summary**:
- The paper introduces an approach that fuses multilingual representations to improve their quality and the downstream performance of multilingual language models in a computationally efficient manner. The work integrates the fusion into low-rank adapters, and comprehensive experiments show that this approach can boost performance across multiple cross-lingual NLU tasks.

**Strengths:**

**Summary Of Strengths**:

- Parameter/computation efficiency: No additional parameters are needed to improve performance, as the fusion occurs within the adapter bottlenecks.
- Quality: The work shows positive experimental results that narrow the gaps between English and non-English language performances across multiple downstream tasks. Meanwhile, the comprehensive discussion and analysis in Section 5 brings insightful ideas to the community regarding this research direction.

**Weaknesses:**

**Summary Of Weaknesses**:
- Limitations: As the authors note, this work focuses on bilingual transfer, so it is very difficult to draw conclusions about the effectiveness of this method in language identification-agnostic scenarios.
- Dependencies on data quality: Multilingual or, especially, low-resource data present significant challenges in LLM training. While this work seems promising, the impact of removing this dependency has not been fully addressed.

**Questions:**

N/A

---

> ### Author Response · Authors · 2024-11-23
> **Author Response to Official Review of Submission11107 by Reviewer xTkb**
>
> > Q1: As the authors note, this work focuses on bilingual transfer, so it is very difficult to draw conclusions about the effectiveness of this method in language identification-agnostic scenarios.
>
> Our evaluation focused on enhancing cross-lingual transfer performance for language models. While language identification may play a role in real-world deployment, it does not impact our findings, as the evaluation settings and conclusions apply consistently across all benchmarked methods, in line with related works [1].
>
> Nevertheless, the NLLB model achieves a 95% F1 score for language identification across 193 FLORES languages, including low-resource languages [2]. This demonstrates that high-accuracy language identification models provide strong performance, supporting the applicability of our FLARE method’s performance benefits in realistic, language identification-agnostic scenarios.
>
> References:
>
> - [1] Huiyun Yang, Huadong Chen, Hao Zhou, and Lei Li. Enhancing cross-lingual transfer by manifold mixup. In International Conference on Learning Representations, 2022. URL https://openreview.net/forum?id=OjPmfr9GkVv.
> - [2] Laurie Burchell, Alexandra Birch, Nikolay Bogoychev, and Kenneth Heafield. 2023. An Open Dataset and Model for Language Identification. In Proceedings of the 61st Annual Meeting of the Association for Computational Linguistics (Volume 2: Short Papers), pages 865–879, Toronto, Canada. Association for Computational Linguistics.
>
> **Actions taken**: We describe the relevance of language identification for the real-world application of FLARE in the Appendix.
>
> > Q2: Dependencies on data quality: Multilingual or, especially, low-resource data present significant challenges in LLM training. While this work seems promising, the impact of removing this dependency has not been fully addressed.
>
> In Section 5, we evaluate the impact of translation quality, including for low-resource languages. Our results demonstrate that FLARE is robust to lower-quality translations, consistently outperforming standard LoRA and other baselines. We agree that challenges associated with low-resource data remain significant in multilingual language models and require further research. We believe that our method provides important contributions toward advancing more equitable language models that are less dependent on high-resource data.
>
> **Given the contributions of the FLARE method and our responses to the reviewer’s remarks, we kindly ask the reviewer to increase the rating score.**

---

> > ### Comment · Reviewer_xTkb · 2024-11-26
> >
> > Thank you for elaborating further. I will keep the score as it is.

---

### Official Review · Reviewer_85DV · 2024-11-05

**Soundness:** 2
**Presentation:** 3
**Contribution:** 2
**Rating:** 3
**Confidence:** 4

**Summary:**

The paper introduces the method FLARE to improve cross-lingual performance in multilingual language models. FLARE utilizes adapters, specifically low-rank LoRA adapters, to fuse representations from source (typically English) and target languages without adding extra parameters or increasing computational complexity. The method reduces the performance gap between English and non-English languages by efficiently combining language-specific information within the adapter layers, demonstrating improvements across tasks such as sentiment analysis and question-answering.

**Strengths:**

- The method is parameter-efficient, as it achieves improved cross-lingual performance without additional parameters.
- The method maintains computational efficiency by fusing representations within adapters rather than extending input sequences.
- The method allows the integration of various representation types, such as latent translations from machine translation models.

**Weaknesses:**

- FLARE’s performance is dependent on the quality of machine translations, potentially limiting its effectiveness in low-resource languages.
- The method has been tested primarily in bilingual scenarios, which may limit its generalizability to more complex multilingual contexts.

**Questions:**

-

---

> ### Author Response · Authors · 2024-11-23
> **Author Response to Official Review of Submission11107 by Reviewer 85DV**
>
> > Q1: FLARE’s performance is dependent on the quality of machine translations, potentially limiting its effectiveness in low-resource languages.
>
> In our evaluation setting, we enrich non-English text inputs with corresponding machine translations to English to increase model performance through our proposed FLARE method. Section 5 explores how translation quality impacts our approach, including cases for low-resource languages. The results show that our method is robust to lower quality machine translations, outperforming regular LoRA and other baselines.
> We comprehensively addressed the quality of machine translations on our method in the manuscript. Furthermore, machine translations enable state-of-the-art cross-lingual transfer performance, particularly on low-resource languages [1].
>
> References:
>
> - [1] Benedikt Ebing and Goran Glavaš. 2024. To Translate or Not to Translate: A Systematic Investigation of Translation-Based Cross-Lingual Transfer to Low-Resource Languages. In Proceedings of the 2024 Conference of the North American Chapter of the Association for Computational Linguistics: Human Language Technologies (Volume 1: Long Papers), pages 5325–5344, Mexico City, Mexico. Association for Computational Linguistics.
>
> > Q2: The method has been tested primarily in bilingual scenarios, which may limit its generalizability to more complex multilingual contexts.
>
> We would appreciate further clarification on what is meant by "more complex multilingual contexts".
> Our evaluation centers on bilingual FLARE adapters, where a separate adapter is trained for each target language. By employing a translate-train approach, extending our method to additional languages requires solely the availability of (machine) translated training data in the new target language. This aligns closely with real-world applications.
>
> As we have comprehensively addressed the reviewer’s concerns in the manuscript, we kindly request that the reviewer re-evaluate the rating score. Given the strengths highlighted, including improved performance without additional parameters, computational efficiency, and flexibility in handling various input representations, we believe a higher rating would be well justified.

---

### Note · Authors · 2024-12-14

I have read and agree with the venue's withdrawal policy on behalf of myself and my co-authors.